# Water Quality and the Effectiveness of European Union Policies

**Yves Steinebach** 

Geschwister Scholl Institute, LMU Munich, Munich 80538, Germany; yves.steinebach@gsi.uni-muenchen.de

**Abstract:** This article is a first attempt to examine the effectiveness of EU water policies in a comparative perspective. It provides a systematic analysis of the relationship between EU water policies and the quality of national water resources for 17 EU member states over a period of 23 years (1990–2012). The analysis reveals that EU policies have contributed to the water quality in the member states. Moreover, it finds that decentralized implementation processes enhance the effectiveness of top-down policy instruments while not making a significant difference for bottom-up policy instruments. Administrative capacities and (neo-)corporatist arrangement seem to play some, yet only minor, role in determining the effectiveness of EU water policies. This way, the article speaks to the literature on EU compliance and implementation and the broader public policy literature.

**Keywords:** European Union; policy; implementation; enforcement; effectiveness; water; instrument choice; bottom-up; top-down

## 1. Introduction

Water is one of the sectors with the most comprehensive coverage by European Union (EU) environmental policy. EU directives have specified emission limit values for water and set standards on how to monitor, report, and manage the water quality of rivers and lakes. The respective provisions triggered considerable changes in national legislative statutes—even in member states with already advanced environmental policy portfolios. Remarkably, however, we know only little about the extent to which the policy changes and actions triggered by the EU level have also achieved their objectives. What is the overall effect of EU policies on the water quality of rivers and lakes in the member states? And why, despite common water protection standards, are some member states better off than others?

To answer these questions, this article provides an encompassing analysis of the relationship between EU water policies and the quality of national water resources. Empirically, the analysis covers 17 EU member states between 1990 and 2012 and focuses on the changes in four crucial water quality indicators (ammonium, nitrates, phosphorus, and dissolved oxygen) over time. The overall logic of the research design is a large-N comparison using standard techniques for the analysis of time-series cross-section (TSCS) data. In essence, the analysis reveals that EU policies have enhanced the water quality in the member states. In addition, it shows that top-down policy instruments benefit from decentralized implementation processes while the effectiveness of bottom-up policy instruments is *not* significantly enhanced when the implementing authorities responsible for water protection are established at the local level. Administrative capacities and (neo-)corporatism seem to play some, yet minor, role in determining the effectiveness of EU water policies.

The contribution of this article is twofold. First, it speaks to the well-established literature on policy compliance and implementation in the context of the EU [1]. While this strand of literature has concentrated on the national impediments to the timely transposition and proper application of EU policies, it has not examined to what extent EU legislation ultimately affects the environmental quality in the member states. Second, this article complements dominant research perspectives in the public

policy and evaluation literature [2,3]. Evaluation studies typically concentrate on individual policies as a unit of analysis. As a consequence, there is often not much variation in the contextual conditions under which policies operate (nor in the policies themselves). This makes it difficult to state whether a causal link found to exist also holds over other settings and to establish reasons why some policies are effective while others are not. The approach chosen in this article, by contrast, takes a broader perspective on the linkage between EU water policies and the quality of national water resources. This way, it allows to identify theoretical patterns that go beyond the scope of single policies.

The article is structured as follows. The first section starts by briefly reviewing the literature on environmental policy effectiveness. Subsequently, Section 2 introduces several theoretical propositions about the connection between environmental policy outputs and outcomes and deduces causal mechanisms from the literature, which then guide the development of the hypotheses. Section 3 turns to the measurement of the dependent and independent variables, before statistically testing the relationship between EU water policy outputs and (changes in) the member states' water quality. The final section summarizes the article's main findings and presents some concluding remarks.

## 2. EU Environmental Policy Effectiveness

The number of scholarly contributions dealing with EU environmental policy in general and EU water policy in particular is impressive (for an overview, see [4]). However, comparative analyses that aim at evaluating and explaining the effectiveness of EU environmental policies are rare and often cover only a limited number of member states and policy measures [5]. Most of the comparative studies dealing with the domestic effects of EU environmental policies have concentrated on the compliance of EU member states and on the national obstacles to smooth policy implementation (see, for instance, [6,7]). The process of policy implementation usually involves (1) the transposition of supranational provisions into national legislation [8,9] and (2) the practical application of the respective provisions by the national implementing authorities [10,11]. Accordingly, policy implementation is the crucial step that puts EU policies into effect. Despite this fact, however, a successful policy implementation does not necessarily and in every instance result in a policy that effectively reduces environmental pollution and degradation. Amongst other aspects, it is possible that the policy instruments chosen by the EU level are generally ill-suited as well as poorly designed to address the underlying policy problem in the first place. Hence, how fast a policy is then transposed into national law or how insistently a policy is executed and enforced becomes irrelevant for the overall outcome. For instance, the EU emission trading scheme (EU ETS), has suffered from severe design flaws that—from the outset—constituted major obstacles to the proper functioning of the policy [12]. Thus, to get a better picture of the actual impact of EU policies on the environmental quality of its member states, we need to put the concept of policy effectiveness at the heart of analysis.

The assessment of policy effectiveness is the central objective of the policy evaluation literature. There are several analyses concerned with the effects of distinct EU water policies (see for instance [13,14]). Typically, evaluation studies seek to separate and 'distil' the effects of a specific policy program or measure from those caused by other policies or factors [15]. To assess these effects as precisely as possible, policy evaluation scholars can choose between different (quantitative) evaluation designs and methods such as randomized control trials, simple before-after, or more advanced matched-pair comparisons [16,17]. All these methods essentially aim at either minimizing the influence of third variables affecting the outcome or at holding all factors but the policy under scrutiny constant. Policy evaluation studies are thus well-suited to produce valid conclusions about the output-outcome nexus for distinct policies, but have difficulties in extrapolating and generalizing the results to other contexts [18,19]. This way, existing evaluation studies have helped to rigorously gauge the effect of individual EU policies but have fallen short in accounting for the variation in environmental policy effectiveness across different temporal and spatial scales. Given the aforesaid shortcomings of the existing policy evaluation literature, the key difference between conventional scientific policy evaluation and the approach of this paper is that it does not attempt to gain knowledge

about a particular policy measure, but seeks to produce more general theoretical insights on EU policy effectiveness and its variation.

## 3. Theorizing the Effectiveness of EU Water Policies

Policy effectiveness can be defined as the degree to which environmental policy measures (*policy outputs*) benefit the environment (*outcomes*). Policy outputs are the immediate result of the decision-making process. Policy outcomes, on the other hand, refer to more general societal or environmental developments that are usually assumed to be at least in part the result of a given set of policy outputs [20,21]. A typical example for an output in the field of environmental governance could be the adoption of new environmental programs or regulations that set out new standards for water and air quality. The corresponding outcomes would then be an increase or decrease of biodiversity in rivers and lakes. A policy can be considered effective if there is a significant relationship between policy outputs and outcomes and if the policy measures taken have a positive influence on the quality of the environment (or a negative influence on environmental pollution).

While this relationship seems to be straightforward, explaining policy effectiveness is actually far from being trivial. The common belief is that the more policy makers do and the more ambitious and the stricter the policy measures taken are, the better the results are going to be. However, one should actually not assume a straight and mechanistic link between means and ends. Rather, even the most ambitious policy can be rendered ineffective by various parameters. Taking this possibility into account, this article does not only ask if, but also under which exact conditions, EU water policies make a real difference for the water quality in the member states. More precisely, it is expected that (1) the policy instruments selected, (2) a country's administrative capacities, (3) the underlying implementation structures, and (4) the system of interest intermediation influences the effectiveness of EU water policies.

### 3.1. Instrument Choice

Over time, the EU has made use of different policy instruments in the area of water policy [22]. In the 1970s and 1980s, numerous water-related directives were adopted. These directives set standards for the quality of water after treatment and for water intended for drinking and bathing. Moreover, they regulated the permissible levels of discharges of particular pollutants. The respective directives mainly applied policy instruments that worked in a 'top-down' manner, i.e., they prescribed obligatory standards that all EU member states had to comply with uniformly. In the late 1980s and early 1990s, the EU shifted its approach from specifying quality and emission standards towards regulating the *processes* of waste water treatment and fertilization by setting infrastructural targets and establishing codes of good agricultural practices [23]. The 2000 Water Framework Directive (WFD), in turn, marks another turning point in the EU's approach toward protecting the environment and improving the water quality in its territory. The directive demands the establishment of the so-called river basins and water bodies that are managed by the respective implementing authorities under the consideration of wider ecological aspects [24]. Unlike previous EU legislation that prescribed detailed objectives that must be achieved, the WFD only specified that all of the designated river basins and water bodies must be of a 'good ecological status'. This way, the WFD has largely relied on the use of 'bottom-up' policy instruments that leave substantial leeway to the member states. In sum, we can thus observe a steady shift from the use of command-and-control (top-down) policy instruments toward more self-regulatory, co-operative, and managerial ones over the history of EU water policy-making [25]. Despite this trend, however, bottom-up policy instruments did not completely replace top-down ones. Rather, the different instruments have been established alongside one another [22].

With regard to the effectiveness of the respective instrument types, different expectations can be formulated. On the one hand, it is possible to argue that top-down interventions are a more appealing policy option as they provide clear stipulations for what both the EU member states and the policy addressees need to do [26]. Member states have to adopt the quality and emission standards prescribed

by the EU and provide the administrative resources necessary for their effective monitoring and enforcement. Policy addressees, in turn, have to comply with the standards or, if not, deal with the chance of getting caught and fined. On the other hand, top-down policy instruments typically apply a 'one-size-fits-all approach' as they do not take account of local peculiarities and differences between the member states. For example, upstream and downstream users of the same river might have to comply with the very same policy standard, but do have quite different capabilities to deal with the underlying environmental problem. This can become particularly problematic once it comes to policy provisions that—at least for the policy addressees in some member states—are difficult if not impossible to comply with [27]. As a result, national authorities might decide not to enforce the policies as strictly as possible or policy addressees deem taking the risk of being sanctioned as the better option—knowing that they will not manage to comply with the policy requirements anyway [28]. From this perspective, bottom-up policy instruments might be equally, if not more, effective than top-down ones given their flexibility to make adjustments in procedure and to adapt to changing contextual conditions [29].

While the existing literature does not univocally point in the direction of generally favoring top-down over bottom-up instruments (or vice versa), it seems reasonable to expect that top-down instruments are associated with higher levels of policy effectiveness. This is due to the fact that most of the bottom-up instruments require the execution of implementation tasks for which national implementers had initially been neither trained for nor staffed appropriately. While national water inspectors were traditionally trained in the technical control of industrial equipment and water quality, the 'new' environmental policy instruments (NEPIs) often involve the long-term planning and management of national water resources [30,31]. It thus seems reasonable to expect that member states are better able to effectively implement top-down rather than bottom-up policy instruments. The respective hypothesis reads as follows:

**Hypothesis 1 (H1).** *The effectiveness of EU water policy is higher when applying top-down rather than bottom-up instruments.*

### 3.2. Administrative Capacities

It is almost a truism that in order to accomplish any kind of objective, policies also need to be adequately implemented at the member state level [32,33]. The capability of national authorities to ensure the target group's compliance with environmental legislation is crucially determined by the quality of administrative structures and practices [6]. The strength of national bureaucracies depends on several aspects [34]. First, it is necessary that public authorities possess the adequate resources required in order to perform the tasks they were established for [35]. In particular, proper environmental enforcement requires sufficient staff, appropriate expertise, and technical equipment. Second, administrations need some autonomy to operate free from undue political influence [36]. Previous studies have shown that governments sometimes have an incentive to sacrifice environmental protection for vested interests and electoral gains—particularly, in times of economic hardship [37,38]. Therefore, they might use their power over the bureaucracy to ensure that the enforcement of a policy goes in their preferred direction [39]. In sum, it is reasonable to expect that the effectiveness of EU water policies varies systematically with a country's administrative resources and the degree of bureaucratic autonomy. The respective hypothesis reads as follows:

**Hypothesis 2 (H2).** *The effectiveness of EU water policies is higher in countries with higher administrative capacities.*

### 3.3. Implementation Structures

A central theme in the organization of implementation processes is how to balance the need for central direction and accountability with the need for local discretion and flexibility. The implementing authorities have substantial decisional discretion during the execution of policies. They may make use of this discretion in cases where their regulatory preferences diverge from those of central policy makers. Similarly, 'bureaucratic' or 'regulatory capture' can lead to gaps between the intended policy goals and the actual outcomes [40]. The terms refer to situations in which the implementing authorities are more beholden to local interest groups or the companies they are supposed to monitor than to their formal mandate [41]. The top-down implementation literature supposes that central (political) control is a potent way to limit bureaucratic drift and that oversight becomes easier with lesser decentralization of the execution of policy programs and the involvement of fewer levels of government in the implementation process [42,43]. The bottom-up perspective, by contrast, posits the exact opposite. Here, it is argued that the implementing authorities must possess substantial procedural discretion to be able to adjust central provisions to local conditions and thus should be established 'closest' to the target group [44,45].

The literature is far from providing a definite answer to the question, *which design of implementation structures delivers better results?* Although, one might hypothesize that decentralized implementation structures involve higher levels of effectiveness in the area of water policy. Water pollution prevention is not (only) a technical or administrative task, but also requires the intense cooperation of various actors, since hazardous effluents cannot be directly reduced at the emission source, but need to be transported through drains and pipes to collective treatment facilities [46]. Therefore, outcomes will heavily depend on the administrators' capabilities to organize themselves in the micro-implementing environment [47]. The respective hypothesis reads as follows:

**Hypothesis 3 (H3).** *The effectiveness of EU water policy is higher when the key implementation authority/authorities is/are located at the regional or local level of government.*

### 3.4. The System of Interest Intermediation

Policy effectiveness is not exclusively influenced by features of the public sector, but also by whether the target groups in the member states are generally inclined and able to comply with the EU provisions made. In this regard, much depends on the relationships between private and public actors as well as between the private actors themselves.

At the heart of these relationships lies the concept of (neo-)corporatism. Corporatism and neo-corporatism have in common that the general form of interaction between the involved actors, both within and across the public and private sector, is cooperative, consensual, and goal-oriented [48]. Given these commonalities of corporatism and neo-corporatism when it comes to the key aspects of interest, one does not have to further distinguish between the two concepts, but can employ the umbrella term (neo-)corporatism in the following discussion. In (neo-)corporatist arrangements, corporations and labor groups are organized in compulsory and non-competitive peak organizations, which hold the representational monopoly for their members vis-à-vis the state [49,50]. Pluralist systems, by contrast, are characterized by a multitude of interest groups struggling to influence policy making by means of lobbyism and 'pressure politics'.

While the benefits of (neo-)corporatism were initially recognized only in the context of macroeconomic issues such as inflation, unemployment, and income policies [51,52], the concept later also emerged as a key factor in explaining policy success and failure in other policy areas [53,54]. Streeck and Kenworthy [55], for instance, argue that in peak organizations, (neo-)corporatist systems must provide special incentives to increase the appeal of associational membership: First, they have to prevent free riding of association members and non-members. Consequentially, they aim at ensuring close monitoring and general compliance with common agreements so that ideally, individual

companies cannot cheat to the disadvantage of others under their authority (see also [56]). This has been particularly acknowledged in the discussion on voluntary environmental agreements where strong branch organizations have demonstrated to possess the capacities to deliver the expected changes from individual member firms [57]. Second, peak organizations may provide membership privileges in the form of material support. This includes, inter alia, legal services, lobbying, and technical advice [58]. Soskice [59] highlights that peak bodies in coordinated market economies play a central role in developing common solutions and fostering the diffusion of 'best practices' throughout the regulated industries. In sum, it is thus reasonable to expect that (neo-)corporatist structures might facilitate policy implementation and thus positively influence the effectiveness of EU policies. Therefore, the hypothesis to be tested is:

**Hypothesis 4 (H4).** *The effectiveness of EU water policy is higher in more neo-corporatist arrangements.*

## 4. Research Design

The empirical analysis focuses on the variation in water quality. The relevant water quality indicators under scrutiny are ammonium, nitrates, and phosphorus as well as dissolved oxygen (DO). Ammonium, nitrates, and phosphorus are pollutants. Accordingly, higher values imply a lower water quality. DO, by contrast, is not a pollutant, but an indication of how well the water can support aquatic plant and animal life. Higher DO values imply a better supply of oxygen and nutrients and thus a higher water quality. In contrast to other environmental outcome indicators such as air pollution, data on water quality is quite difficult to obtain and compare. This is mainly due to the fact that the data on water quality are not aggregated on national levels, but instead measured in the countries' main rivers and lakes. Unfortunately, the available OECD dataset does not consistently cover the *same* rivers over time [60]. The analysis thus refers to the average water quality across all rivers for each country–year and pollutant (for a comparable approach, see [61]). All pollutant levels are standardized using 1990 as the base year for total pollution (100 percent). DO is rescaled so that lower values indicate a better environmental policy performance. If the year 1990 is missing, the first year in the time series is chosen as the reference year. This procedure allows to pool the different pollutants into one single dependent variable (clustered in country-years and pollutants) and thus to test for the overall effect of EU policies across different pollutants. A benefit of the OECD data is that the water quality is measured at the downstream frontiers of the rivers. This ensures that the respective data refer to the respective country.

The analysis covers all EU member states included in the OECD dataset on the quality of national water basins with the exception of Portugal and Greece for which there is a complete lack of data on some pollutants. The sample includes Austria, Belgium, Czech Republic, Denmark, Finland, France, Germany, Great Britain, Hungary, Ireland, Italy, Latvia, Netherlands, Poland, Spain, Slovakia, and Sweden. The countries under scrutiny vary along theoretically relevant dimensions. Among others, the selected countries differ substantially with regard to (1) the quality of the state's bureaucracy, (2) the implementation structures in the area of water policy enforcement, and (3) the system of interest mediation. The investigation period ranges from 1990 to 2012. Unfortunately, there is no data available from the year 2012 onwards. Yet, Steinebach and Knill [38] identified a significant drop in (the production of) EU environmental policy proposals by the Commission in the years following the financial and economic crisis (see also [62]). In other words, even when having a longer time series for the dependent variable, there would not be many EU water policies to test for the output-outcome-nexus. Accordingly, a focus on the data and time frame available seems to be sufficient to address this paper's research question. Some of the countries under analysis have not been member of the EU in the year 1990, but only joined the EU during the time period. This, however, does not present major difficulties for the analysis as candidate countries are also expected to comply with EU's environmental acquis before accession [63].

### 4.1. The Independent Variable: Assessing Environmental Policy Outputs

To assess environmental policies and their ambitiousness, the analysis relies on a concept proposed by Knill et al. [64], (see also [65]). Knill et al. [64] assess the ambitiousness of policy outputs by the evaluation of three distinct components: changes in policy targets, policy instruments, and instrument settings (comprising of both instrument level and scope).

Policy targets refer to the specific objectives addressed in a certain policy field and thus focus on the question, *who or what exactly is regulated by the legislators?* In the area of water policy, this might be ammonium emissions from industrial discharges into continental surfaces water for instance. Policy instruments are an indicator of which specific tools, out of a range of options, are used by policy makers in order to achieve their targets. It thus refers to the question, *how the policy makers intend to solve its environmental problems?* A specific policy target is often addressed by the use of various instruments. The instrument level, in turn, refers to the exact calibration of a policy instrument. For instance, in the case of an obligatory emission standard, the level prescribes the ammonium value for lead emissions in surface water, e.g., 0.02 ppm. The instrument scope, in turn, covers the specific cases or addressees targeted by a certain policy instrument. It can prescribe, for example, whether a regulation relates to all industrial plants or only to those that apply for an operating license for the first time. Figure 1 presents the different policy components and how they are tied together. Every policy target has at least one instrument, which in turn has a level and a scope.

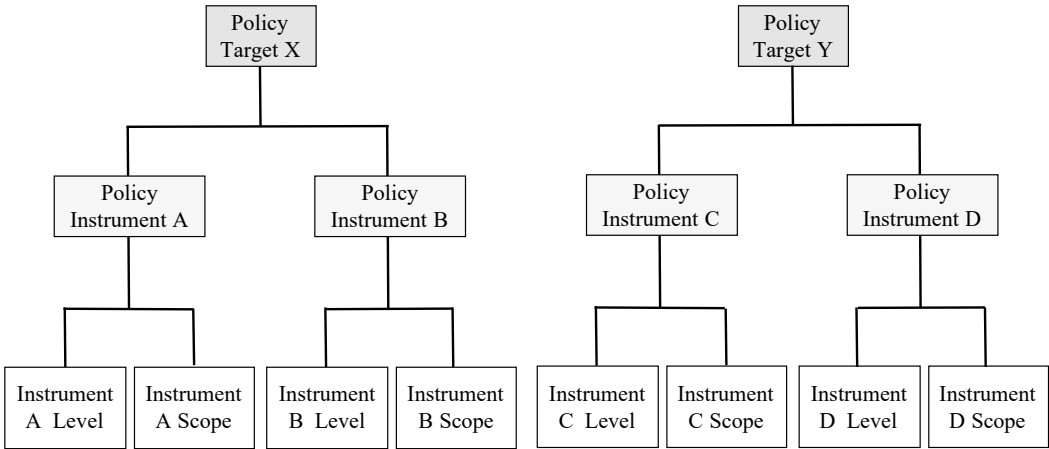

**Figure 1.** Policy components and their connections.

As a result of the presented hierarchical structure among the components, changes can be weighted differently by a simple logic of aggregation. By definition, changes in policy targets have to involve changes in at least one policy instrument and its calibration (value "4"). Following the same logic, the establishment of a new policy instrument inevitably leads to changes of an instrument's level and scope (value "3"). By contrast, both level (value "1") and scope (value "1") (or both) may change without any implication for the existence of either policy targets or instruments.

To capture the policy targets and instruments in the area of water policy, the data collection process was done by using a list of predefined policy targets and instruments that could be derived from the existing literature. A list of the policy targets and instruments under scrutiny is provided in Tables 1 and 2. In the outcome dimension, the selected policies are expected to affect (1) the respective pollutant they address as well as (2) the DO value. The presented concept and coding were applied to all EU legislations being adopted during the investigation period (see Table A1 in the Appendix A). In a second step, it was checked whether or not and, if so, when the respective policy measures have entered into force (as opposed to the adoption date) in the member states consulting EUR-Lex as well as reports from the European Commission. This procedure allowed to (1) check for the extent to which member states had to change their existing provisions in the area of water policy in response

to EU legislations and (2) identify cases of non-compliance, i.e., when member states did not or only insufficiently incorporate specific EU measures into their national policy corpus.

**Table 1.** Water Policy Targets.

| Water Protection Policy |
|---|
| 1.    Ammonium in continental surface water |
| 2.    Dissolved oxygen (DO) in continental surface water |
| 3.    Nitrate ($NO_3^-$) in continental surface water |
| 4.    Phosphorus in continental surface water |
| 5.    Ammonium from industrial discharges into continental surface water |
| 6.    Nitrate ($NO_3^-$) from industrial discharges into continental surface water |
| 7.    Phosphorus from industrial discharges into continental surface water |

**Table 2.** Environmental Policy Instruments.

| Instrument | Description |
|---|---|
| Obligatory standard | A legally enforceable numerical standard, typically involving a measurement unit, e.g., mg/L |
| Prohibition/ban | A total or partial prohibition/ban on certain emissions, activities, products, etc. |
| Technological prescription | A measure prescribing the use of a specific technology or process |
| Planning instrument | A measure defining areas or times that deserve particular protection |
| Voluntary measures | Voluntary agreements or commitments between the state and private actors or by private actors alone |
| Information-based instrument | Information provided by the state or the polluters indicating the environmental externalities of a certain product or activity |

## 4.2. The Determinants of Policy Effectiveness

Given this paper's interest in the varying degrees of policy effectiveness of different *instrument types*, the analysis broadly distinguishes between 'top-down' and 'bottom-up' policy instruments. In this context, obligatory standards, prohibition/ban, and technological prescriptions represent top-down policy instruments. Planning, voluntary, and information-based instruments, in turn, are considered bottom-up policy instruments. Planning instruments include both the designation of areas requiring particular attention as well as the development of environmental action plans. It must be noted, however, that there is no commonly accepted *positive* definition of bottom-up policy instruments. Rather, bottom-up policy instruments must be considered a catch-all category for instruments that have not been used traditionally to target water pollution and do not require the compliance with uniform and centrally set standards.

To measure a country's *administrative capacities*, the World Bank's Worldwide Governance Indicators (WGI) project is used as a broad proxy for bureaucratic capacity. This indicator combines several variables that measure the "quality of public services, the quality of the civil service and the degree of its independence from political pressures, the quality of policy formulation and implementation, and the credibility of the government's commitment to such policies" [66]. It is a fairly standard measure when interested in the role of the administration in policy transposition and enforcement [67] and encompasses a number of variables, which have been previously employed to capture administrative

capacity such as the International Country Risk Guide's 'Bureaucracy Quality' indicator [68,69]. In the absence of data on policy or sector-specific implementation capacities, this comprehensive indicator appears to be the best available choice to measure bureaucratic capacities. A potential weakness of the WGI might be that the indicated values are normalized with zero mean and unit standard deviation. This leads to a global mean of zero for government effectiveness for each time period across all countries in the sample and an overall limited range of possible values. As a result, countries could be slightly better (or worse) off simply due to changes of government effectiveness in other countries. Although this could cause some problems when engaging in comparisons over time, this challenge is of limited relevance for the analysis given that our overall country sample is of rather homogenous nature, i.e., only consists of EU member states and thus advanced capitalist democracies.

To assess the design of the *implementation structures*, it was coded whether the key implementing authorities in the area of water policy are established at the central (federal) (0) or the regional (local) level (1) using a simple binary dummy variable. Here, Spain constitutes a hybrid case since central watershed agencies are in charge of monitoring water quality targets as well as inspecting industrial facilities. These agencies are headed by a president who is nominated by the Spanish minister of the environment [70]. Yet, if a river runs entirely within the territory of an autonomous community, the regional water administrations are in charge of managing these water resources. In the context of this paper, the focus is on the federal agencies as all of the selected rivers that are used in the OECD dataset on water quality do cross through more than one of the autonomous communities [56]. Moreover, it needs to be mentioned that some of the countries under scrutiny shifted the implementation responsibilities during the investigation period from the local to the central level (or vice versa). Amongst others, this has been due to the first-time establishment of central environmental protection agencies or the administrative reforms required in the context of the WFD. The data on implementation structures were gathered from various sources. These were primarily the information made available online by the responsible authorities. In addition, other sources such as academic contributions and the reports of international organizations (in particular, the OECD and the European Energy Agency (EEA)) were consulted.

A country's system of *interest intermediation* is usually captured by referring to the distinction between (neo-)corporatism and pluralism. Despite the concepts' ubiquity in comparative politics, most attempts to empirically model and operationalize corporatism have remained merely cross-sectional in scope [71,72]. Although Hicks and Kenworthy [73] and Siaroff [74] constructed indices of corporatism for various points in time (usually one per decade), the actual temporal variance is very limited. A particularly useful solution is offered by Jahn's corporatism index [75]. This index covers both the structural and functional features of corporatist arrangements, and the extent to which the economy is encompassed by the agreements they made. The respective data are readily accessible online and cover all countries under scrutiny as well as the entire investigation period.

### 4.3. Control Variables

Apart from EU water policies, their design, and the way policies are executed and enforced, several other factors may affect changes in water quality. As a consequence, one needs to test the relationship between water policy outputs and outcomes against several control variables. First and foremost, it might be the case that there are national policy measures in place alongside the provisions induced by the EU level. Given the EU's strong influence in regulatory matters, however, these additional measures are largely restricted to market-based incentives such as water pollution taxes or effluent fees [76]. Thus, the per capita revenue from water and wastewater taxation are included in the analysis. These data are readily available, provided by the OECD [77].

The 'Environmental Kuznets Curve' hypothesis posits that environmental pollution and economic growth relate to each other in a nonlinear fashion. It assumes that while environmental damages initially increase with countries' economic development, they reach a peak point before ultimately decreasing [78]. Thus, the potential influence on changes in water quality must be accounted for

by incorporating the natural log of GDP per capita in the analysis (OECD 2018). However, it is not only the absolute level of economic development that may affect the water quality. Water pollution from industrial production and activities also strongly fluctuates with times of economic upturns and downturns. Accordingly, the analysis takes account of these short-term changes by including the GDP growth rate in analysis [79]. In addition, there might be confounding effects related to demographic changes. Consequentially, and in accordance with previous studies, changes in the urban population are included in the analysis. This aspect is controlled by the share of the population living in urban areas [80]. The dependent and independent variables are summarized in Table A2 in the Appendix A.

### 4.4. The Analytical Model

The relationship between policy outputs and outcomes is estimated by means of standard techniques for the analysis of time-series cross-section data. As customary in the literature, all independent variables are lagged by one year. The only exceptions are the levels of and changes in countries' economic output (GDP pc and GDP growth), which are assumed to exert an immediate effect on the water quality. By contrast, they were opted for a more inductive approach in specifying the lag structure between water policy outputs and outcomes. Given that there is no prior knowledge on *when*—if at all—water policy outputs make a difference for the outcome, it seems reasonable to allow the time lags to vary by using the R-square value as the efficiency criterion. In the case of new technological standards for industrial plants, for instance, it may take years until a sufficiently high number of industrial plants has been equipped with the new technologies before a true difference is noticeable and significant. It is virtually impossible to specify these lagged effects deductively for all cause-effect relationships in the model ex ante. Depending on the exact model, it took about two years (after entering into force at the member state level) that the policies under scrutiny made a difference in the quality of national waters. This lag structure also ensures that the policy output changes *antecede* (possible) changes in the outcome dimensions, thus reducing issues of reversed causality.

The standard errors are corrected to account for clusters in the data structure. Here, the Driscoll and Kraay's covariance matrix estimator is used to correct for standard errors [81]. Although Driscoll and Kraay's standard errors tend to be slightly more 'optimistic' than other estimators, given our specific case, they tend to produce more robust results than other approaches as the cross-sectional dimension (N) is relatively large compared to the temporal one (T) [81]. Moreover, there is a certain risk that regression disturbances are not only correlated over time, but also between the different spatial units, due to (1) potential trade-offs in simultaneously reducing different water pollutants and (2) the possibility of plant reallocations to other countries to lower production cost.

A substantial share of the unit-specific variation is already eliminated by using 1990 as the base year. However, given that the dependent variable does not only encompass different countries, but also different water quality indicators (the three pollutants plus DO), there still might be some unobserved factors that generally (dis)favor outcome changes which are not yet covered by the analysis. For instance, it be might the case that some pollutants generally have a higher reduction potential and are thus simply more prone to changes than others. To control for this unobserved unit-level heterogeneity and to ensure that it is not specific countries or indicators driving the regression results, dummy variables for each country and pollutant (ammonium, DO, nitrates, and phosphorus) are applied. Moreover, year dummies are used to control for time dynamics.

When testing for the assumptions of linear regression, the analysis revealed that the model suffers from non-normality and heteroscedasticity (non-constant variance). To rectify this problem, it was necessary to transform the dependent variable. Here, a square root transformation of the dependent variables was used to address the respective problems. A Box–Cox transformation (best lambda: ~0.434) yielded comparable results.

### 4.5. Examining the Effectiveness of EU Water Policies

Figure 2 presents the results for the effect of EU policy outputs on the water quality in the member states. Smaller values imply *less* water pollution and hence *better* water quality. The figure shows that the effect of both top-down and bottom-up policies is negative and statistically significant. This finding suggests that EU policies contribute to a reduction of water pollution in the member states. Yet, there are no significant differences between the two instrument types. If at all, it seems that bottom-up policy instruments exert a stronger influence on the dependent variable as the respective coefficient tends to be (even) more negative. Moreover, the uncertainty involved is slightly smaller as indicated by the shorter confidence bands. As a result, hypothesis H1 cannot be confirmed.

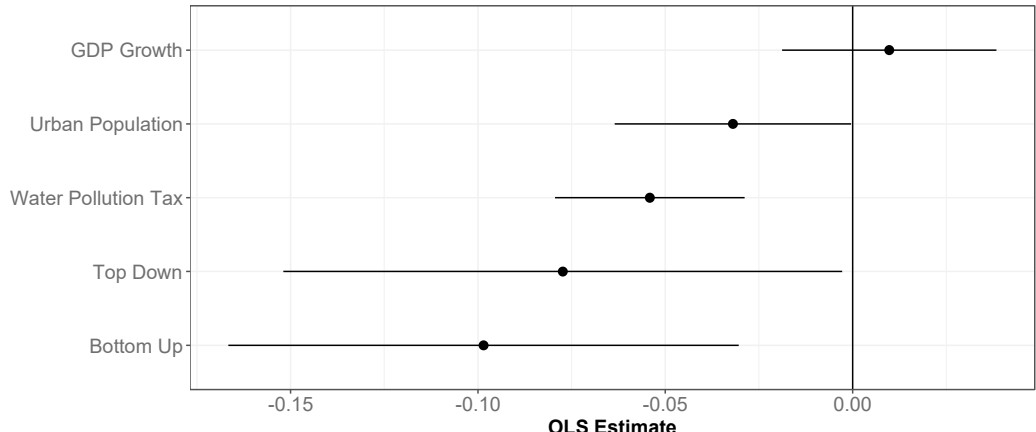

**Figure 2.** Determinants of water pollution, 1990–2012. Note: Point estimates and 90% confidence intervals (CI).

But how strong is the effect of EU top-down and bottom-up water policies exactly? This question is difficult to answer given that the dependent variable had to be transformed. The coefficients and their associated results only apply on the scales on which they were estimated. It is thus not possible to give a clear and easily interpretable number (such as percentages) on how strong EU policies have benefitted water quality in Europe when being compared with the 1990 baseline values. A possible solution to this is to compare the coefficients of the top-down and the bottom-up variable with those of another (control) variable. Figure 2 suggests that the least ambitious change of EU water policies (a change of an instrument's setting or scope; value "1") has about the same effect as if each and everybody in a country has to pay two more dollars per year for polluting water. Across all countries and years in our sample, this would imply a quite remarkable average increase in water pollution tax returns of about 16 percent (for a discussion on the effect of varying levels of pollution taxes, see [82]). Given this paper's approach in measuring policy ambitiousness, the introduction of a completely new policy instrument (value "3") even has a three times stronger effect.

The previous analysis has shown that EU policies do generally make a difference in the quality of national water resources. This is remarkable as previous large-N studies assessing the effectiveness of environmental policies have not found a straight and unconditional link between policy outputs and outcomes [64]. Despite this insight, however, we do not know whether the effectiveness of EU water policies is always and everywhere the same or whether it varies by the context in which the respective policies are implemented. Figure 3 presents the interaction effects between (1) the two instrument types and (2) the different factors hypothesized to affect the proper functioning of water policies (for the full regression table, please consult Tables A3 and A4 in the Appendix A).

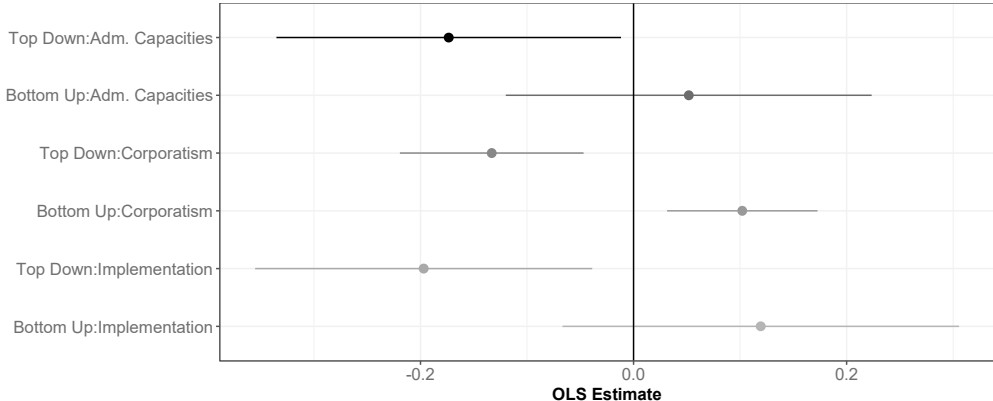

**Figure 3.** Determinants of water policy effectiveness, 1990–2012. Note: Point estimates and 90% confidence intervals.

The first aspect that stands out when consulting Figure 3 is that top-down and bottom-up policy instruments seem to differ in the conditions they need for leveraging their full potential. A recurrent argument in the literature on policy design and instrument choice is that not all policies require the same amount of implementation capacities to become effective. In this context, it has been argued that more hierarchical forms of governmental intervention often require more encompassing implementation capacities due to their need for constant inspection and control [20,83]. This is also somewhat reflected by this analysis. The analysis shows that higher administrative capacities in the member states are associated with more effective top-down policy instruments, while not making a true difference for bottom-up instruments.

Yet, continuous-by-continuous interactions are complex and difficult to interpret as the effects of both constituent variables always depend on the level of the other variable. In this context, marginal effect plots can facilitate interpretation. Figure 4 displays how the effect of water policies on the outcome dimension changes with different levels of administrative capacities. The marginal effects are calculated based on the full model including time trends as well as country- and pollutant-fixed effects. The figure reveals that top-down policies do *not* lead to better environmental outcomes when bureaucratic quality is deficient. Yet, we cannot definitely conclude that the opposite scenario is true—i.e., that higher levels of administrative capacities also increase the effectiveness of EU water policies—as the confidence bands do still include the zero line. Accordingly, we cannot confirm hypothesis 2 (H2) that the effectiveness of EU water policies is higher in countries with higher administrative capacities. Rather, it seems that a bare minimum of administrative capacities is necessary that the water policies can, but not necessarily must, make a difference in the quality of national water resources.

A similar observation can be made with regard to the relationship between the system of interest intermediation and the effectiveness of EU top-down policies (Figure 5). Here, again, it is only for low values of the (neo-)corporatism index that we can safely conclude that policies are ineffective. The opposite, in turn, does not apply to higher index values. For bottom-up policies, this relationship is even reversed—without, however, reaching levels of statistical significance for high and low values of the (neo-)corporatism index (Figure 6). Accordingly, we cannot confirm hypothesis 4 (H4) that EU water policies are more effective in neo-corporatist arrangements—neither for top-down nor bottom-up policy instruments.

The most interesting finding of the analysis is that top-down policy instruments seem to benefit from decentralized policy implementation while *not* making a significant difference for bottom-up instruments (see again Figure 3). Here, the initial theoretical expectation was that administrators in decentralized implementation structures find it *generally* easier to adjust central policy provisions to local peculiarities and thus ensure higher levels of policy effectiveness. A possible yet inconclusive answer to this puzzling finding could be that bottom-up instruments do allow per se for substantial discretion to adjust them to the local context. Top-down policies, by contrast, must be implemented

decentrally and in intense cooperation with local actors to avoid a rigid 'one-size-fits-all approach', thereby exploiting the policies' full potential. In sum, we can hence confirm hypothesis 3 (H3) for top-down policy instruments, but not for bottom-up ones.

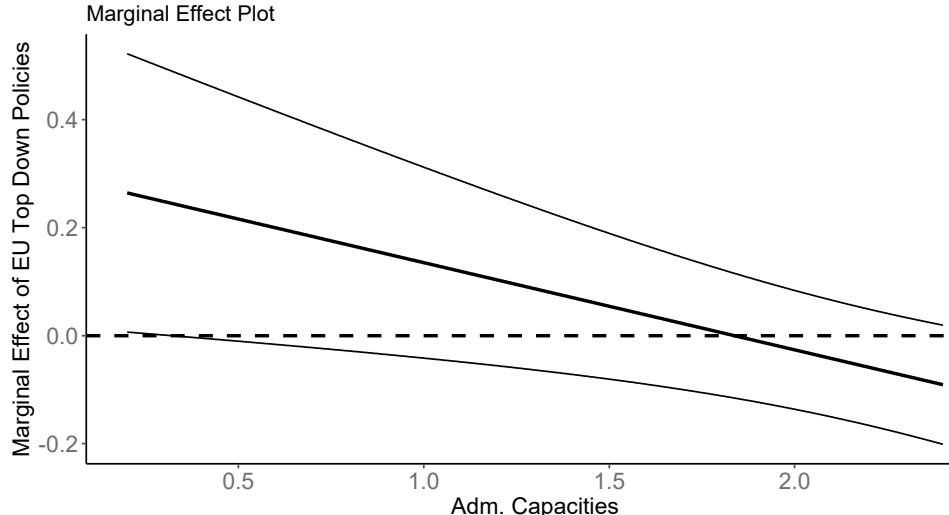

**Figure 4.** Marginal effect of EU top-down policies over different levels of administrative capacities (with 90% CI).

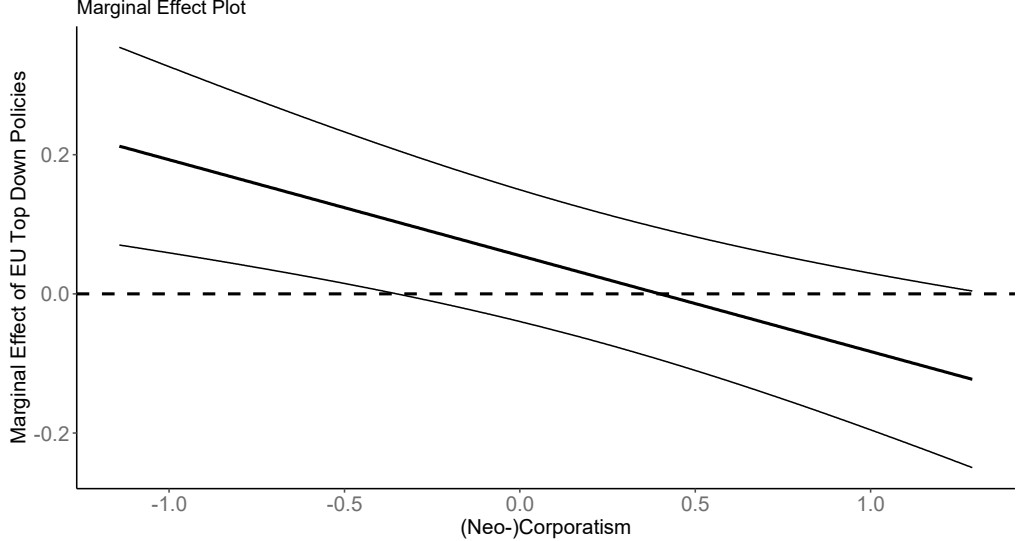

**Figure 5.** Marginal effect of EU top-down policies over different levels of (neo-) corporatism (with 90% CI).

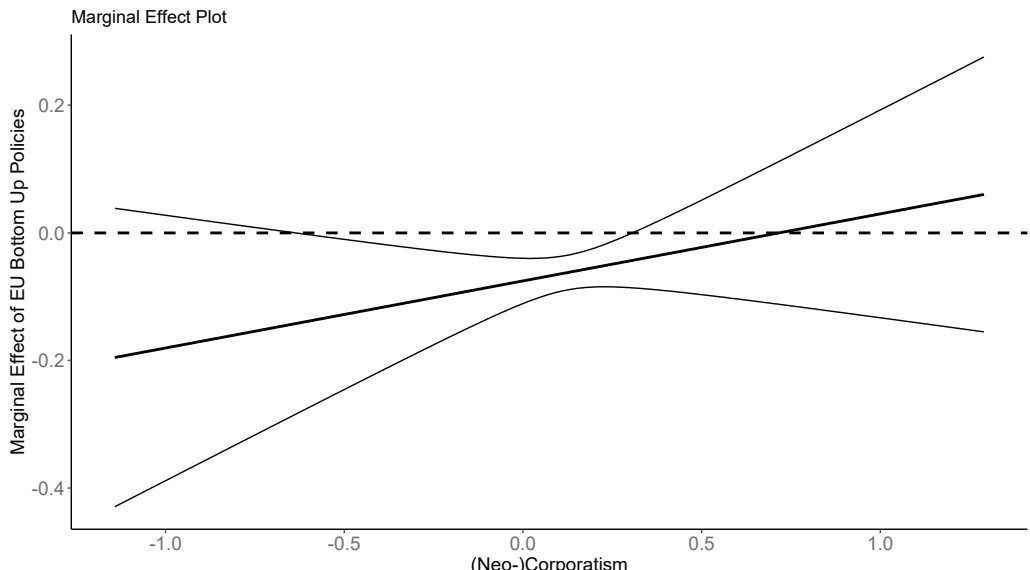

**Figure 6.** Marginal effect of EU bottom-up policies over different levels of (neo-) corporatism (with 90% CI).

## 5. Discussion

The first aspect that stands out when taking a look at the above findings is that most of the factors under consideration do somewhat contribute to the proper functioning of top-down policy instruments, while having no or the exact opposite effect on bottom-up policy instruments. A possible explanation for this could be that the analysis itself takes on a top-down perspective in approaching the implementation process. Essentially, it is presumed that they are only a few of the dominant factors that condition the implementation effectiveness. The influence of these factors then varies, if at all, by the instrument type applied rather than from policy to policy or due to local circumstances. Accordingly, one might get a very different picture when shifting the level of analysis from country or sectoral to the policy level.

Another insight from the above analysis is that administrative capacities and (neo-)corporatism play some, yet minor, role in determining the effectiveness of EU water policies. More precisely, the analysis revealed that in case of (very) low levels of administrative capacities and (neo-)corporatism, top-down policies will definitely not make a difference for the outcome dimension. Conversely, however, this does not imply that higher levels of administrative capacities and (neo-)corporatism are also necessarily associated with higher degrees of policy effectiveness. This finding is remarkable as the countries' administrative capacities have frequently been found to be a crucial factor when it comes to the timely transposition and proper application of EU policies [1,67]. A potential explanation for this result could be that public administrations have some, but only limited, influence on whether the target group will ultimately change its behavior—and this even when the government actually possesses the capabilities to adequately monitor and enforce public policies.

In regard to the existing literature, the findings concerning the implementation structures are in so far remarkable as previous studies have argued that the appropriate design of the implementation structures depends on the underlying policy problem (and the degree of policy uncertainty and political conflict involved) [47,84]. This paper's analysis, in turn, has shown that the 'right' design of the implementation structures does not only depend on the policy problem at hand, but also on the instrument type applied.

## 6. Conclusions

This article examined the relationship between EU water policies and the quality of national water resources. By means of quantitative analysis techniques, it was detected that EU policies have

helped to improve the water quality of member states' rivers. Moreover, the analysis revealed that the effectiveness of water policies is not the same everywhere in Europe, but varies by the national implementation context. Here, in particular, the design of the implementation structures was found to make a crucial difference depending on the instrument type (top-down versus bottom-up) used.

So how is it possible to further enhance the effectiveness of EU water policies? First and foremost, it is necessary to highlight that EU water policies do (already) make a quite good job in tackling water pollution in Europe. Accordingly, there is only limited room for improvement. The biggest potential for further enhancing the effectiveness of EU water policies lies in improving national implementation capacities and adjusting them to the requirements of the respective instrument type applied. Accordingly, member states must overcome how things were traditionally done and, if necessary, re-design the underlying implementation structures in response to the type of policy instrument prescribed by supranational legislation. The reforms required in the context of the WFD have shown that institutional change is indeed difficult due to the persistence of national administrations, but far from being impossible (see [85]).

All in all, this paper must be considered a first attempt to examine the effectiveness of EU water policies in a comparative perspective. The analysis showed that it might be promising for future research to examine more thoroughly how different types of policy instruments and determinants located at the implementation stage interact. In this context, it might be promising to check, for instance, the effect of coordination demands (if more than one public authority is involved) and the involvement of non-state actors. Moreover, future studies might have a look on the impact of a country's administrative tradition (legalistic versus managerial) on the functioning of the different instrument types.

**Funding:** This research received no external funding.

**Conflicts of Interest:** The author declares no conflict of interest.

**Appendix A**

**Table A1.** List of EU legislation in the Area of Water Policy.

| EU Legislation | | Brief Description |
|---|---|---|
| COUNCIL DIRECTIVE 91/271/EEC concerning urban waste-water treatment | (1) | MS must ensure that all agglomerations are provided with waste water collecting and treatment system. MS must ensure that, while treatment, pollutants are reduced to a certain extent (*oblig. standard*) |
| COUNCIL DIRECTIVE 91/676/EEC concerning the protection of waters against pollution caused by nitrates from agricultural sources | (1) | MS must establish codes of good agricultural practice, to be implemented by farmers on a voluntary basis (*voluntary instrument*) |
| | (2) | MS must establish action programs in respect of "designated vulnerable zones" (*planning instrument*) |
| COUNCIL DIRECTIVE 96/61/EC concerning Integrated Pollution Prevention and Control (IPPC) | (1) | MS shall take the necessary measures to provide that installations are applying the 'best available techniques' (BAT) (*tech. prescriptions*) |
| DIRECTIVE 2000/60/EC establishing a framework for Community action in the field of water policy | (1) | MS are required to designate river basins and prepare river basin management plan (*planning instrument*) |
| REGULATION (EC) No 166/2006 concerning the establishment of a European Pollutant Release and Transfer Register and amending Council Directives 91/689/EEC and 96/61/EEC | (1) | MS have to provide easily accessible key environmental data from industrial facilities (*information-based instruments*) |

**Table A2.** Descriptive Statistics.

| Variables | N | Mean | SD | Min | Max |
|---|---|---|---|---|---|
| Water Pollution (*percentages, 1990 baseline*) | 1195 | 85.82 | 53.82 | 0.10 | 834.30 |
| Water Pollution (after square roots transformation) (*percentages, 1990 baseline*) | 1541 | 8.93 | 2.48 | 0.32 | 28.88 |
| Top-Down Policy Instruments | 1541 | 0.15 | 0.61 | 0 | 3 |
| Bottom-Up Policy Instruments | 1541 | 0.34 | 1.07 | 0 | 6 |
| GDP pc | 1470 | 31,087.23 | 9811.66 | 8283.83 | 48,746.62 |
| GDP Growth | 1536 | 2.38 | 3.16 | −14.40 | 11.90 |
| Urban Population (*percentages of total population*) | 1564 | 73.30 | 10.80 | 54.30 | 97.70 |
| Water Pollution Taxes (*per capita revenue from water and wastewater taxation*) | 1243 | 12.65 | 24.75 | 0.00 | 111.40 |
| Administrative Capacities | 952 | 1.42 | 0.57 | 0.20 | 2.40 |
| Implementation Structures | 1564 | 0.63 | 0.48 | 0.00 | 1.00 |
| (Neo-)corporatism | 1424 | 84.87 | 49.30 | −1.14 | 1.29 |

**Table A3.** Determinants of Water pollution, 1990–2012 (main models).

| Independent Variable | Model | | |
|---|---|---|---|
| | (1) | (2) | (3) |
| Top-Down Policies | −0.077 * (0.045) | −0.077 * (0.045) | −0.077 * (0.045) |
| Bottom-Up Policies | −0.098 ** (0.041) | −0.098 ** (0.041) | −0.098 ** (0.041) |
| GDP pc (logged) | 3.272 *** (0.614) | 3.272 *** (0.614) | 3.272 *** (0.614) |
| GDP Growth | 0.010 (0.017) | 0.010 (0.017) | 0.010 (0.017) |
| Urban Population | −0.054 *** (0.015) | −0.054 *** (0.015) | −0.054 *** (0.015) |
| Water Pollution Tax | −0.031 * (0.019) | −0.031 * (0.019) | −0.031 * (0.019) |
| $R^2$ | 0.44 | 0.44 | 0.44 |
| N | 956 | 956 | 956 |
| Country | 17 | 17 | 17 |
| Time Trend | YES | YES | YES |
| Country FE | YES | YES | YES |
| Pollutant FE | YES | YES | YES |

*** $p < 0.01$, ** $p < 0.05$, * $p < 0.10$; model 2 controls for the influence of a country's administrative tradition; model 3 controls for the influence of the worlds of compliance.

**Table A4.** Determinants of Water pollution, 1990–2012 (main models).

| Independent Variable | Model | | |
|---|---|---|---|
| | (1) | (2) | (3) |
| Top-Down Policies | 0.322 * | 0.157 ** | 0.057 |
| | (0.148) | (0.079) | (0.043) |
| Bottom-Up Policies | −0.079 | −0.123 * | −0.077 |
| | (0.165) | (0.073) | (0.068) |
| GDP pc (logged) | 3.393 ** | 3.254 *** | 3.584 *** |
| | (0.637) | (0.618) | (0.819) |
| GDP Growth | −0.010 | 0.004 | −0.009 |
| | (0.015) | (0.019) | (0.016) |
| Urban Population | −0.054 | −0.049 *** | −0.024 |
| | (0.016) | (0.016) | (0.027) |
| Water Pollution Tax | −0.034 | −0.022 * | −0.018 |
| | (0.031) | (0.016) | (0.019) |
| Adm. Capacities | −0.207 | | |
| | (0.312) | | |
| Impl. Structures | | −0.073 | |
| | | (0.138) | |
| (Neo-)corporatism | | | −0.603 |
| | | | (0.958) |
| Top-Down Policies: Adm. Capacities: | −0.173 * | | |
| | (0.098) | | |
| Bottom-Up Policies: Adm. Capacities: | 0.052 | | |
| | (0.104) | | |
| Top-Down Policies: Impl. Structures | | −0.197 ** | |
| | | (0.096) | |
| Bottom-Up Policies: Impl. Structures | | 0.119 | |
| | | (0.113) | |
| Top-Down Policies: (Neo-)corporatism | | | −0.133 ** |
| | | | (0.052) |
| Bottom-Up Policies: (Neo-)corporatism | | | 0.102 ** |
| | | | (0.042) |
| $R^2$ | 0.47 | 0.45 | 0.45 |
| N | 891 | 891 | 891 |
| Country | 17 | 17 | 17 |
| Time Trend | YES | YES | YES |
| Country FE | YES | YES | YES |
| Pollutant FE | YES | YES | YES |

*** $p < 0.01$, ** $p < 0.05$, * $p < 0.10$.

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
