# Peer review of "Water Quality and the Effectiveness of European Union Policies"

_water, doi:10.3390/w11112244_

Round 1
Reviewer 1 Report
I think this paper focuses on a very important topic--water quality and the effectiveness of EU policies. The abstract states the article contributes to literature on EU compliance and implementation and the broader public policy and evaluation literature. I believe this paper does achieve this, however I think the Conclusions section of this paper could be greatly expanded especially since the paper provides a lot of detail in the Introduction, EU Environmental policy Effectiveness, Theorizing the Effectiveness of EU Policies, Research Design, Examining the Effectiveness of EU Water Policies sections.
The paper does not have a Discussion section and the Conclusions section is only 2 paragraphs and does not expand upon the main conclusions that have been made from the extensive analysis done by the authors or it does not provide next steps for enhancing the effectiveness of EU policies and improving the quality of national water resources from the research undertaken. This paper could greatly benefit from a discussion section and a more informative conclusions section.
Also, on page 7 lines 242-244 there is not a good explanation why the analysis covered the 17 EU member states and not other states? I was not clear why these 17 EU member states were chosen.
I noticed some spelling mistakes in the document (e.g. analysis was spelled incorrectly) on page 7 line 274. Please make sure to do a thorough spell check.
Starting on page 8 line 314 and throughout the rest of the document, one of the authors refers to himself/herself as "I". See page 9, lines 323, 341, 348, page 10 line 409 etc. I thought this paper was written by 2 authors not just one?? Using "I" is distracting for the reader, I think these sentences need to be reworded to take out "I".
There is a lot of information in the Annex, however, it did not seem as though all of the tables were cited in the text of the document. Please check this.
Also, shouldn't the Annex appear after the References section?
Author Response
Before going into detail on the changes made to our manuscript, we want to use the opportunity to thank the two reviewers for their extensive and thoughtful comments on my paper. We think that the quality of the reviews has helped to improve the paper/the analysis.
Remark 1
The paper does not have a Discussion section and the Conclusions section is only 2 paragraphs and does not expand upon the main conclusions that have been made from the extensive analysis done by the authors or it does not provide next steps for enhancing the effectiveness of EU policies and improving the quality of national water resources from the research undertaken. This paper could greatly benefit from a discussion section and a more informative conclusions section.
Answer
Very true. We now added a discussion section plus a more informative conclusion.
Remark 2
Also, on page 7 lines 242-244 there is not a good explanation why the analysis covered the 17 EU member states and not other states? I was not clear why these 17 EU member states were chosen.
Answer
There is actually not much of a strategy behind the case selection as we are using the full sample of EU countries for which the OECD provides data on the water quality (with the exemption if Portugal and Greece for which a substantial share of data is missing). We now openly admit this. Yet, it is still true that the 17 member scrutiny do vary along theoretically relevant dimensions (so that one might talk about a diverse case selection strategy).
Remark 2
I noticed some spelling mistakes in the document (e.g. analysis was spelled incorrectly) on page 7 line 274. Please make sure to do a thorough spell check.
Answer
I corrected the mistakes and made another spelling ceck.
Remark 3
Starting on page 8 line 314 and throughout the rest of the document, one of the authors refers to himself/herself as "I". See page 9, lines 323, 341, 348, page 10 line 409 etc. I thought this paper was written by 2 authors not just one?? Using "I" is distracting for the reader, I think these sentences need to be reworded to take out "I".
Answer
The article is now entirely written in the third person.
Remark 5
Also, shouldn't the Annex appear after the References section?
Answer
True. We changed this.
Reviewer 2 Report
The analysis is conducted on a small number of year. The data set is old (data only till 2012). The data series should be updated to include data till 2018.
The regression models are not valid in the presented forms. Some coefficients are not significant. The assumptions on errors were not checked (errors homoskedasticity, normal distribution, independence). The overall approach for all 17 EU countries is not the best choice. Those aggregated indicators might not be relevant. The author should divide the sample of countries in clusters having countries with similar characteristics in the same cluster and the estimation should be made separately for each cluster.
More comparisons with expectations and with previous studies are required.
The abstract should be extend as to highlight the proper results and their significance for literature.
The author should add more recent references from literature.
Author Response
Before going into detail on the changes made to our manuscript, we want to use the opportunity to thank the two reviewers for their extensive and thoughtful comments on my paper. We think that the quality of the reviews has helped to improve the paper/the analysis.
Remark 1
The analysis is conducted on a small number of year. The data set is old (data only till 2012). The data series should be updated to include data till 2018.
Answer
Unfortunately the data set as provided by the OECD ends in 2012 so that it cannot be extended. We now reflect on this caveat in the text.
Remark 2
The regression models are not valid in the presented forms. The assumptions on errors were not checked (errors homoscedasticity, normal distribution, independence).
Answer
This is true and a very valid point. When checking for these aspects, it turned out that the model suffers from non-normality and heteroscedasticity. We addressed both issues by a square roots transformation of the dependent variable (a box cox transformation yielded comparable results). Moreover, we removed the control variable 'industrial share' as the inclusion of the variable led to non-independence/auto correlation in the model (most probably due to the consecutive NA at the end of the time series). Both aspect - but in particular the transformation - somewhat affected our result. The analysis is thus entirely renewed/rewritten.
Remark 3
The overall approach for all 17 EU countries is not the best choice. Those aggregated indicators might not be relevant. The author should divide the sample of countries in clusters having countries with similar characteristics in the same cluster and the estimation should be made separately for each cluster.
Answer
We do not get 100 percent the reviewers point here. Is s/he asking to subset the data? The problem with subsetting the data is that predictors might turn insignificant simply due to a (much) lower number of observations. Moreover, it is also not the paper's analytical interest to look for the EU water policy effect in, lets say, Southern, Central, Northern Europe etc. We think that the country-fixed effect approach (as done in paper) takes account of the fact that there might be some (unobserved) cross-country difference. In the conclusion we now suggest that a potential way for future research might be to examine the impact of a country’s administrative tradition (legalistic versus managerial) on the proper functioning of the different instrument types.
Remark 4
More comparisons with expectations and with previous studies are required.
Answer
This is now done in an extra discussion section.
Remark 5
The abstract should be extend as to highlight the proper results and their significance for literature.
Answer
Given the new results, the abstract had to be rewritten anyways. We now try to report the findings more accurately. A longer/extended abstract was not possible due to the restrictions made by the journal.
Remark 5
The author should add more recent references from literature.
Answer
This was done where possible.
Round 2
Reviewer 2 Report
The paper could be published if authors describe more the methods with their practical significance for this application.
Author Response
Remark
The paper could be published if authors describe more the methods with their practical significance for this application.
Answer
We are not a 100 percent sure about the reviewer's concern. We interpreted it as being more explicit on the paper's practical significance with regard to the method (here the DV transformation) and our concept for measuring policy change. On page 24 we thus added the following paragraph.
But how strong is the effect of EU top-down and bottom-up water policies exactly? This question is difficult to answer given that the dependent variable had to be transformed. The coefficients and their associated results only apply on the scales on which they were estimated. It is thus not possible to give a clear and easily interpretable number (such as percentages) on how strong EU policies have benefitted water quality in Europe when being compared with the 1990 baseline values. A possible solution to this is to compare the coefficients of the top-down and the bottom-up variable with those of another (control) variable. Figure 2 suggests that the least ambitious change of EU water policies (a change of an instrument’s setting or scope; value “1”) has about the same effect as if each and everybody in a country has to pay another two dollars per year for polluting water. Across all countries and years in our sample, this would imply a quite remarkable average increase in water pollution tax returns of about 16 percent (for a discussion on the effect of varying levels of pollution taxes see Guoet al.2018). Given this paper’s approach in measuring policy ambitiousness, the introduction of a completely new policy instrument (value “3”) has even a three times stronger effect.
